# Effects of Combined Upper and Lower Limb Plyometric Training Interventions on Physical Fitness in Athletes: A Systematic Review with Meta-Analysis

**DOI:** 10.3390/ijerph20010482

**Published:** 2022-12-28

**Authors:** Nuannuan Deng, Kim Geok Soh, Zeinab Zaremohzzabieh, Borhannudin Abdullah, Kamariah Md Salleh, Dandan Huang

**Affiliations:** 1Department of Sports Studies, Faculty of Educational Studies, Universiti Putra Malaysia, Serdang 43400, Selangor, Malaysia; 2Institute for Social Science Studies, Universiti Putra Malaysia, Serdang 43400, Selangor, Malaysia; 3School of Educational Studies, Universiti Sains Malaysia, Gelugor 11800, Penang, Malaysia; 4College of Physical Education, Chong Qing University, Chongqing 400044, China

**Keywords:** plyometric training, power, sprint, strength, athletes

## Abstract

Objective: We aimed to meta-analyze the effects of combined upper and lower limb plyometric training (ULLPT) on physical fitness attributes in athletes. Methods: A systematic literature search was conducted in Web of Science, SPORTDiscus, PubMed, and SCOPUS, for up to 13 August 2022. Controlled studies with baseline and follow-up measures were included if they examined the effects of ULLPT on at least one measure of physical fitness indices in athletes. A random effects meta-analysis was performed using the Comprehensive Meta-Analysis software. Results: Fifteen moderate-to-high-quality studies with 523 participants aged 12–22.4 years were included in the analyses. Small to large (ES = 0.42–1.66; *p* = 0.004 to <0.001) effects were noted for upper and lower body muscle power, linear sprint speed, upper and lower body muscle strength, agility, and flexibility, while no significant effects on static and dynamic balance were noted (ES = 0.44–0.10; all *p* > 0.05). Athletes’ sex, age, and training program variables had no modulator role on the effects of ULLPT in available data sets. Conclusions: ULLPT induces distinct neuro-muscular adaptations in the upper and lower body musculature and is an efficient method for enhancing athletes’ physical fitness.

## 1. Introduction

The ultimate objective of sports training in competitive sports is to develop excellent sports performance, and athletes’ physical fitness is the most fundamental aspect of improving athletic ability [1]. Higher levels of physical fitness (e.g., strength, jump ability) are required to perform sport-specific tasks successfully [2,3,4]. For instance, to achieve better stroke performance in tennis, players will require a combination of power, agility, sprint, and well-developed aerobic fitness [5]. On the other hand, athletes with excellent physical fitness can increase their own sports ability and avoid injuries [6,7], and also have self-awareness of whether they can withstand higher-intensity sports training or competition [8]. Notably, it is well known that strength training can assist athletes to get in better physical condition [9,10,11]. Plyometric training is widely recognized as a practical option for participants to achieve better physical performance among the numerous available strength exercises.

Conceptually, plyometric training involves an eccentric contraction followed by an explosive concentric contraction [12,13]. Meanwhile, researchers frequently refer to the transition between the eccentric to the concentric phases of action as the stretch-shortening cycle (SSC) [12,13]. The physiological principles of plyometric training include two models: (1) in the mechanical model, the series elastic component of the muscle is stretched and stores elastic energy like a spring during eccentric muscle movements; if the muscles immediately initiate the concentric movement, the stored energy is released and contributes to the total force output [14,15]. (2) The neurophysiological model postulates that the muscle stretch caused by the eccentric action stimulates the muscle spindle, which triggers the stretch reflex; this stretch reflex enhances the agonist’s ability to generate more force during the subsequent concentric contraction of the muscle [15,16,17]. Moreover, these two models are integrated into three distinct phases of plyometric exercises, including (1) the eccentric phase: the muscle spindles are stimulated, and the elastic energy is stored [17]; (2) the amortization phase: the activated muscle spindles stimulate spinal motor neurons, which excite muscle fibers [17,18]; (3) the concentric phase: this last phase is the consequence of the two models interacting, producing forceful explosive concentric contractions [17,19]. Previous reviews have extensively discussed the potential of these changes to improve athletes’ competitive performance [20,21,22].

Plyometric training often involves skipping, jumping, and hopping [23]. These exercises have been shown to improve jumping ability [24], sprinting [25], and running economy [26]. However, in addition to plyometric workouts for the lower extremities, the development of upper extremity exercises has been ongoing, and related benefits have been seen, such as improved tennis maximal serve speed and throwing velocity of baseball players [27,28]. Closed kinetic chain exercises (e.g., push-ups) and open kinetic chain medicine ball exercises (e.g., ball throws) are forms of upper body plyometrics [29,30]. Athletes in many sports must utilize both their upper and lower bodies extensively. For example, basketball players’ lower limbs are involved in sprinting, jumping, and faking motions, whereas shooting, rebounding, and blocking need upper limb strength and skills [31]. Based on the aforementioned theories, neuro-mechanical adaptations induced by ULLPT may improve the efficiency of SSC in the upper and lower body muscle tissue, leading to superior force output [12,13,14,15,16,17,18]. Therefore, it seems reasonable to hypothesize that the ULLPT program may have some potential and theoretical training advantages for athletes’ upper and lower limbs.

Currently, an increasing proportion of experimental research has explored the effect of ULLPT on athletes’ physical fitness attributes. For instance, in Hammami et al.’s [32] study, ULLPT is a time-efficient and valuable tool for increasing whole-body physical performance in young female handball players. Nevertheless, all of this evidence has yet to be compiled and analyzed. On the other hand, there have been multiple reviews and meta-analyses published that examine the effect of lower limb plyometrics on various physical fitness indices [33,34,35,36]. Additionally, upper-body plyometric training on physical performance was also summarized [37]. To address the lack of systematic reviews and meta-analyses on this knowledge, comprehensive research is necessary for exploring this field. Thus, this study aimed to explore the effects of ULLPT vs. controls on various measures of physical fitness (e.g., upper and lower body muscle power/strength) relevant to athletes. Furthermore, we were inquisitive about the impact of potential moderators on intervention effects, including subject-related variables and training-related variables.

## 2. Materials and Methods

The present study followed the 2020 PRISMA guidelines [38], and the review protocol has been registered on Inplasy.com [INPLASY202290059], (accessed on 14 September 2022).

### 2.1. Literature Search

To obtain relevant publications on our topic, we searched four electronic databases on 13 August 2022: Web of Science, SPORTDiscus, PubMed, and SCOPUS. The Boolean search syntax shown below was used: “plyometric training” OR “plyometric exercise*” OR “stretch-shortening cycle” OR “stretch-shortening exercise*” AND “player*” OR “athlete*” OR “sportsman*” OR “sportswoman*” OR “sportsperson*”. Moreover, a search on Google Scholar was conducted to avoid missing any potential articles that met the criteria. Furthermore, the reference lists of selected papers and reviews were checked to determine whether additional acceptable works may be included in this research. We also consulted with librarians throughout the entire search process. The detailed primary databases search strategy is presented in Appendix A.

### 2.2. Eligibility Criteria

Studies included in this review had to provide descriptive information about PICOS (population, intervention, comparator, outcome, and study design). The selection criteria applied in this study are summarized in Table 1.

According to the establishment time of the databases, the last seventy-seven years of original, full-text studies (1945–2022) published in peer-review journals were accepted for this review. Excluded were books, book chapters, other cross-sectional studies, and articles not concentrating on ULLPT (e.g., solely upper limb plyometric exercises or lower limb plyometrics). Also excluded were prospective studies, retrospective studies, publications of which only an abstract was made accessible, special communications, case reports, invited commentaries, letters to the editor, patents, and overtraining studies. Studies on detraining were considered for inclusion if they consisted of a training period before a detraining period. Finally, given the challenges of translating articles and previous research has demonstrated that nearly all of the literature (99.6%) on plyometric jump training was in English [39], we limited our evaluation to works originally published in English.

### 2.3. Study Selection and Data Collection Process

In order to select the studies that would be included in our study, two authors (DN, HD) used specialized software and manual selection to filter the retrieved duplicate articles (EndNote X9 for Windows, Clarivate Analytics). Following that, the same authors independently read all article titles and abstracts to eliminate irrelevant literature before analyzing relevant full-text papers. If the two reviewers disagreed, a third author (KGS) was consulted until a consensus was reached. The authors recorded the reasons for exclusion when excluding articles with full text.

Two reviewers (DN, HD) extracted the required data from collected papers using a Microsoft Excel spreadsheet (Microsoft Corporation, Redmond, WA, USA), and a third author (KGS) double-checked the data to make sure everything was accurate.

### 2.4. Data Items

The following categories of physical fitness indices were included, but not limited to:Muscle power (e.g., medicine ball throw (MBT), countermovement jump without (CMJ) or with arm wing (CMJa));Muscle strength (e.g., chess/leg press);Linear sprint speed (e.g., 10 m);Agility (e.g., Illinois Test);Flexibility (sit and reach test);Balance (static/dynamic).

Aside from the data items listed above, adverse effects were recorded, and descriptive information of subject characteristics (e.g., sample size, gender) and the ULLPT interventions (e.g., duration, rest time) were also contained.

### 2.5. Methodological Quality Assessment

The PEDro scale was adopted to evaluate the research’s qualities of methodology. This scale is perhaps the one that is utilized the most commonly in the literature on plyometric training [34,40,41]. Moreover, it is consistent with other scales, such as the Cochrane risk of bias instrument [42]. The quality assessment was generally interpreted as ≤3 = poor, 4–5 = moderate, and 6–10 = high. These scores were used if the methodological quality of the publication had been evaluated and mentioned in the PEDro database (or similar topics of published articles). Two reviewers independently assessed the methodological quality of all included publications (DN and HD). If the two reviewers disagreed on the score, it was discussed with a third author (KGS) until a consensus was reached.

### 2.6. Summary Measures

Following previous relevant research [22,35], three or more studies providing baseline and follow-up data for the same parameter were meta-analyzed. Based on pre- and post-intervention performance means and standard deviations, between-group effect sizes (ES; Hedge’s g) were computed (SD). The data were standardized using the post-score SD. Meta-analyses used the inverse-variance random-effects model because it allocates proportionate weights to trials based on their standard errors [43] and accounts for heterogeneity across studies [44]. If the needed information was not available in the original article or additional materials, the authors were contacted. If contact was not returned or data were unavailable, the paper was not included in the meta-analysis. The ES values are displayed alongside 95% confidence intervals (CIs), and were interpreted as follows: trivial, <0.2; small, 0.2–0.6; moderate, >0.6–1.2; large, >1.2–2.0; very large, >2.0–4.0; extremely large, >4.0 [45]. For studies with multiple ULLPT groups, the control group was often split into equal halves to facilitate inter-group comparisons [46]. The Comprehensive Meta-Analysis software (version 3; Biostat, Englewood, NJ, USA) was used to analyze all of the available data.

### 2.7. Synthesis of Results and Risk of Bias across Studies

Heterogeneity was analyzed with the use of the *I*^2^ statistic. It was established that 25%, 25–75%, and >75% represent low, moderate, and high levels of heterogeneity, respectively [47]. A meta-analysis of the risk of bias in individual research was performed using an extended version of Egger’s test [48]. When Egger’s test result was statistically significant (*p* < 0.05), a sensitivity analysis was also conducted.

### 2.8. Additional Analyses

Subgroup analyses were carried out in order to investigate the potential effects that moderator factors may have. We employed a random-effects model, and we pre-selected pertinent sources of heterogeneity that could affect the training effects via the authors’ discussion and the incorporation of study characteristics: program duration (weeks), training frequency (sessions per week), the total number of training sessions, the time of ULLPT session itself (main part), and intensity of training. Participants were divided using a median split [49,50] for training length (i.e., ≤7 vs. >7 weeks), training frequency (i.e., ≤2 vs. >2 sessions per week), the total number of ULLPT sessions (i.e., ≤14 vs. >14 sessions), and the time of ULLPT session itself (main part) (i.e., <30 vs. ≥30 min), the median was calculated if at least 3 studies provided data for a given moderator. Moreover, the gender (female vs. male) and age (≥15 vs. <15 years old) of the athletes have also been considered to be moderator factors. The median split strategy was used to separate the participants [49,50]. Each of these factors was stratified in a meta-analysis, with a *p* < 0.05 deemed the statistical significance level.

## 3. Results

### 3.1. Study Selection

The database search yielded 2870 articles (531 from PubMed, 1124 from SCOPUS, 661 from SPORTDiscus, and 615 from Web of Science), plus 7 more from Google Scholar and reference lists. After removing duplicates, 1470 research papers remained, 1185 studies were discarded via title and abstract reading, and two researchers reviewed 285 full-text publications for eligibility. In all, 16 RCTs met all of the inclusion/exclusion criteria and were reviewed. The entire selection procedure is depicted in Figure 1.

### 3.2. Methodological Quality of the Included Studies

One of the 16 selected papers was considered of a lower quality and therefore excluded [51]. Of the 15 papers, 4 scored four-to-five points (moderate quality), and 11 scored six to seven points (high quality) (Table 2). The PEDro median score across studies was 5.8. The research’ methodological quality was thus rated as moderate to high.

### 3.3. Population Characteristics

Table 3 contains detailed information on the athletes in the experimental groups. (1) Athlete categorization. The 15 studies recruited swimmers [54], tennis players [55], volleyball players [57,62], soccer players [58], gymnasts [59], judokas [61,64], handball players [32,63], and basketball players [31,52]; (2) Sample size, gender, and age. There were 523 total participants, including 262 females and 261 males. Participants in eligible studies ranged in age from 12 to 22.4 years.

### 3.4. Intervention Characteristics

Table 4 summarizes the characteristics of the ULLPT interventions for the fifteen studies. (1) In terms of the exercise type, ULLPT training for the upper extremities includes medicine ball exercises, chess presses, and push-ups, while ULLPT training for the lower extremities includes variations of bounding, hopping, skipping, and jumping drills. However, the three studies did not describe the exercises in-depth [54,61,64]; (2) The duration of the fifteen studies ranges from 4 to 16 weeks, with one to three weekly training frequency sessions.

### 3.5. Meta-Analysis Results

Appendix A shows the data that were used for the meta-analyses.

#### 3.5.1. Effect of ULLPT on Upper and Lower Body Muscle Power

A significant, moderate effect of ULLPT was found on upper body muscle power (i.e., MBT) (ES = 0.87; 95% CI = 0.48–1.26; *p* < 0.001; *I*^2^ = 51.7%; Egger’s test *p* = 0.899; *n* = 223; Figure 2A). The relative weight of each study in the analyses varied between 8.61% and 15.44%.

A significant, moderate to large effect of ULLPT was noted on lower body muscle power (vertical jump height), which comprised CMJ (ES = 0.82; 95% CI = 0.52–1.13; *p* < 0.001; *I*^2^ = 43.5%; Egger’s test *p* = 0.392; *n* = 316; Figure 2B), CMJa (ES = 1.06; 95% CI = 0.32–1.79; *p* = 0.005; *I*^2^ = 84.4%; Egger’s test *p* = 0.203, *n* = 219; Figure 2C) and squat jump height performance (ES = 1.16; 95% CI = 0.74–1.57; *p* < 0.001; *I*^2^ = 23.8%; Egger’s test *p* = 0.491, *n* = 136; Figure 2D). The relative weight of each study in the analyses varied between 6.33% and 28.77%.

#### 3.5.2. Effect of ULLPT on Linear Sprint Speed

A significant, large effect of ULLPT was noted on linear sprint speed, which comprised the 5 m linear sprint (ES = 1.26; 95% CI = 0.44–2.07; *p* = 0.003; *I*^2^ = 71.0%; Egger’s test *p* = 0.326; *n* = 98; Figure 3A), 20 m linear sprint (ES = 1.50; 95% CI = 0.82–2.17; *p* < 0.001; *I*^2^ = 59.0%; Egger’s test *p* = 0.330; *n* = 105; Figure 3B), and 30 m linear sprint performance (ES = 1.66; 95% CI = 0.97–1.74; *p* < 0.001; *I*^2^ = 78.4%; Egger’s test *p* = 0.447; *n* = 214; Figure 3C). The relative weight of each study in the analysis ranged from 11.78% to 37.29%.

#### 3.5.3. Effect of ULLPT on Upper and Lower Body Muscle Strength

A significant, moderate effect of ULLPT was observed on upper body muscle strength (e.g., handgrip) (ES = 0.63; 95% CI = 0.07–1.25; *p* = 0.048; *I*^2^ = 77.7%; Egger’s test *p* = 0.686; *n* = 198; Figure 4A). The relative weight of each study in the analysis ranged from 12.57% to 15.83%. Moreover, a significant, small effect of ULLPT was noted on lower body muscle strength (e.g., leg press) (ES = 0.42; 95% CI = 0.07–0.77; *p* = 0.019; *I*^2^ = 0.00%; Egger’s test *p* = 0.330; *n* = 124; Figure 4B). The relative weight of each study in the analysis ranged from 20.0% to 30.29%.

#### 3.5.4. Effect of ULLPT on Agility

A significant, large effect of ULLPT was noted on agility (ES = 1.31; 95% CI = 0.64–1.99; *p* < 0.001; *I*^2^ = 79.6%; Egger’s test *p* = 0.103; *n* = 205; Figure 5). The relative weight of each study in the analysis ranged from 14.8% to 18.1%.

#### 3.5.5. Effect of ULLPT on Flexibility

A significant, small effect of ULLPT was observed on flexibility performance (ES = 0.48; 95% CI = 0.15–0.81; *p* = 0.004; *I*^2^ = 0.00%; Egger’s test *p* = 0.139; *n* = 143; Figure 6). The relative weight of each study in the analysis ranged from 9.1% to 22.0%.

#### 3.5.6. Effect of ULLPT on Balance

A non-significant, trivial, small effect of ULLPT was noted on static balance performance (ES = 0.44; 95% CI = −0.01–0.90; *p* = 0.056; *I*^2^ = 46.6%; Egger’s test *p* = 0.780; *n* = 141; Figure 7A) and dynamic balance performance (ES = 0.10; 95% CI = −0.29–0.49; *p* = 0.622; *I*^2^ = 00.0%; Egger’s test *p* = 0.880; *n* = 101; Figure 7B). The relative weight of each study in the analysis ranged from 6.9% to 13.3%.

### 3.6. Additional Meta-Analyses

Due to a restricted number of trials (three per moderator), only 28 analyses of moderators were performed (as shown below).

Regarding subject-related moderator variables, when compared to younger athletes, no significant increases were observed following ULLPT in their older counterparts, for CMJa (>15 years of age, ES = 0.74; ≤15 years of age, ES = 1.52; *p* = 0.273), and CMJ (>15 years of age, ES = 0.93; ≤15 years of age, ES = 0.73; *p* = 0.550), nor for upper body muscle strength (>15 years of age, ES = 0.84; ≤15 years of age, ES = 0.89; *p* = 0.914). Similarly, between females and males, no significant improvements were seen following ULLPT, for CMJa (females, ES = 1.37; males, ES = 0.63; *p* = 0.343) and CMJ (females, ES = 0.79; males, ES = 0.88; *p* = 0.795), upper body muscle strength (females, ES = 0.92; males, ES = 0.78; *p* = 0.754), nor for 30 m linear sprint time (females, ES = 2.07; males, ES = 1.21; *p* = 0.206), and flexibility (females, ES = 0.41; males, ES = 0.57; *p* = 0.650).

Regarding training-related moderator variables, no significant improvements were observed following ULLPT when athletes performed >8 weeks (and >16 total ULLPT sessions), as opposed to when they performed ≤8 weeks (and ≤16 total ULLPT sessions), for CMJa (former, ES = 1.38; latter, ES = 0.64; *p* = 0.306), and CMJ (former, ES = 0.97; latter, ES = 0.70; *p* = 0.390), as well as for upper body muscle strength (former, ES = 1.34; latter, ES = 0.53; *p* = 0.013), 30 m linear sprint time (former, ES = 2.37; latter, ES = 1.15; *p* = 0.076), agility (former, ES = 1.52; latter, ES = 1.12; *p* = 0.600), and flexibility (former, ES = 0.34; latter, ES = 0.71; *p* = 0.287).

### 3.7. Adverse Effects

For studies included in the review, none reported soreness, fatigue, pain, damage, injury, or adverse effects related to the ULLPT intervention.

## 4. Discussion

### 4.1. Effect of ULLPT on Upper and Lower Body Muscle Power

We examined the effect of ULLPT on MBT performance for upper body muscle power, and there were significant moderate (ES = 0.87) benefits following ULLPT. Of note, MBT is the most well-known and extensively utilized indirect test for assessing upper-limb muscle power in sports [65]. Briefly, MBT variations are probably the result of upper body adaptations, such as increased motor unit activation and neuronal firing frequency, or simply by cognitive–motor learning effects [66,67]. Additionally, more than half of the ULLPT protocols in our review were designed with medicine ball exercises (e.g., medicine ball throwing, medicine ball chest pass), which comprise one of the most commonly used methods to improve upper body muscle power and are used in the development of upper limb plyometric exercises [68]. Meanwhile, Ignjatovic et al. [69] stated that the ability to accurately simulate forceful moves that are needed for handball success makes medicine ball workouts an excellent training activity for young female handball players. The findings of various studies involving athletes point to the potential use of upper limb plyometric exercise in improving upper-body muscular power. For instance, Palao et al. [70] found that professional female volleyball players who engaged in upper-body plyometric exercises dramatically boosted their overhead medicine ball throws after eight weeks of training. Similar conclusions were made by Singla and Hussain [71], who revealed that eight weeks of medicine ball plyometric training dramatically improved cricket players’ upper body power. In this review, six studies assessed upper body muscle power among young and adult athletes. Five of six studies achieved significant improvement, however, in a study done by Sadeghi et al. [54] on young male swimmers, researchers discovered no significant change between pre-and post-tests of seated medicine ball throwing following six weeks of ULLPT sessions. One potential explanation for the lack of impact on the medicine ball throwing test is that the training regimen for this study was mostly focused on the lower limbs; another possibility is that the individuals’ skills are a major factor in the testing of their performance Sadeghi et al. [54].

On the lower body muscle power aspect, we examined the effect of ULLPT on vertical jump height (i.e., CMJa, CMJ, SJ) and found that there were significant moderate-to-large gains following ULLPT for CMJa (ES = 1.06) and CMJ (ES = 0.82), as well as for squat jump height (ES = 1.16). Normally, the vertical jump has been the most commonly and consistently used test for assessing athletes’ lower limb power output [72]. The propulsion of the lower limbs during a vertical jump is regarded to be a useful indicator of leg muscle power in athletes [73]. An improvement in vertical jump height might be crucial for trained players in jumping-intensive sports such as volleyball [74]. Specifically, the vertical jump is an essential component of volleyball players’ spike, block, topspin, and floating serves [75]. In short, plyometric training-induced changes in jumping performance were caused by several neuromuscular adaptations, such as (1) enhanced neural drive of agonist’s muscles, along with (2) changes in muscle–tendon mechanical stiffness features, (3) changes in muscle size and/or architecture, and (4) alterations in single-fiber mechanics [65,76,77,78,79]. Besides, because the ULLPT incorporates different jumping exercises, it is not surprising that this type of movement improves dramatically. Interestingly, motivation was considered to have a crucial impact on vertical jump performance [80], and plyometrics (e.g., depth jumps) could be seen as a motivating tool [81]. Moreover, our meta-analytic results align with previous research that revealed plyometric jump training had moderate effects (ES = 0.49) on CMJ performance in individual-sport athletes [22], small-to-moderate effects (ES = 0.56–0.80) on three vertical jump performances (i.e., CMJ, CMJa, SJ) among amateur and professional volleyball players [35], and small-to-large effects (ES = 0.44–3.59) on vertical jump height (CMJ, SJ, DJ) in female athletes [24]. Although numerous papers on plyometric training have shown enhancements in vertical jump height, many researchers have failed to report significant positive effects of this kind of training on vertical jump height [82,83], and some have even noted negative effects [84]. These contradictory findings might be explained by small sample sizes [80] or discrepancies in training regimens [74].

According to our findings, ULLPT seems to be a practical approach for increasing athletes’ upper and lower body muscular power. This factor should be considered in future experimental investigations.

### 4.2. Effect of ULLPT on Upper and Lower Body Muscle Strength

Our meta-analysis showed significant changes in upper body muscle strength (ES = 0.87, moderate). On the other hand, the observed improvement in lower body strength (ES = 0.42), which was small but meaningful, reached a statistically significant threshold. A recent meta-analysis [35] and other studies [85,86] have shown increases in lower body muscle strength performance of athletes after plyometric jump training, and the current results show that if the regimen includes upper and lower limb training, both upper and lower extremity benefits can be obtained. The changes might be attributed to several reasons. First, the nature of the training protocol, the type of plyometric exercises (i.e., combining upper and lower limb exercises) used, and second, some neural adaptations are consistent with those discussed previously in the muscle power section [12,79,87]. Additionally, the reduction of antagonist’s muscles and the improvement of co-activation of synergist muscles could be another important reason for this part of the changes [88]. In the present review, six studies assessed muscle strength in basketball, judo, handball, and tennis athletes, and positive gains were reported in these results [31,32,53,55,61,64]. Sharma and Multani [53] observed that two weeks of ULLPT improved calf strength in adult basketball players, but not sufficiently enough to enhance grip, leg, or back strength; all strength measures improved statistically after four weeks. Behringer et al. [55] concluded that young tennis players with greater increases in l0RM values (e.g., leg press, chest press) during the eight-week ULLPT intervention period had greater gains in maximal serve velocity. Of note, when compared for the experimental and control groups, Uzun and Karakoc [61] found that ten weeks of ULLPT significantly improved back strength in adult male judokas, however, there was no significant difference in hand grip and leg strength; the reasons for the lack of significant enhancement in these measures are not clear. Kurniawan et al. [64] conducted an eight-week intervention for adult male judo athletes and found that the ULLPT groups with active recovery or passive recovery had significant change effects on the hand grip strength, but there were no significant differences in hand grip strength when compared to the control group. Meanwhile, plyometric training has been shown to improve this fitness factor in athletes from a variety of different sports, and there is a significant body of research to back this claim. For example, Ioannides et al. [89] report that six weeks of plyometric exercises improve muscular strength performance in young karate athletes. According to Fathi et al. [90], 16 weeks of plyometric exercises contributed to greater improvements in strength performance in adolescent volleyball players. Therefore, our findings provide evidence that ULLPT is an appropriate method to improve both upper-and lower-extremity muscle strength performance among athletes.

### 4.3. Effect of ULLPT on Linear Sprint Speed

Our meta-analysis showed significant large (ES = 1.26–1.53) improvements in linear sprint speed (5 m, 20 m, and 30 m) after the ULLPT program. Similar gains in athletes’ linear sprint performance in our findings were congruent with the reports of the previous plyometric jump training meta-analysis on individual-sport athletes, but they showed small (ES = 0.23) improvements [22]. Several physiological factors could explain increases in sprint performance, and these mechanisms are similar to the previously discussed improvement in lower body muscle power and strength. First, plyometric exercise alters motor unit recruitment patterns (primarily in the fast muscle fibers) [89]. It raises the frequency of motor unit activation, which enhances maximum muscular strength and power capacity in the lower extremities, enabling athletes to burst more quickly at the beginning of sprints and achieve longer stride lengths [12,91,92]. Next, a systematic review has highlighted that combining vertical and horizontal jumping exercises is the key to improving sprinting performance [25,93]. A considerable number of the ULLPT regimens in our review included a combination of horizontal and vertical jumps. Regarding this, these jump drills in ULLPT program can be an excellent strategy to help different kinds of athletes improve their sprinting abilities. Additionally, 1 out of 15 studies in our review discovered that changes over shorter distances (5 and 10 m) were not statistically significant following ULLPT in young female handball players [32]. Kotzamanidis [94] and Meylan and Malatesta [23] stated that improving the initial sprint acceleration (over 5 and 10 m) was more difficult than enhancing the maximum speed, which they explained could be due to the smaller margin for improvement and the different forces involved. However, the study by Hammami et al. [63] noted opposite results in 5 m and 10 m sprint times, and the data showed significant gains after a ten-week ULLPT program in young female handball players. Differences in plyometric training regimens (e.g., duration, frequency) may assist in understanding the varying magnitudes of sprint speed changes observed across research [20]. Notably, one study [58] included in the review compared the effects of short-term (6 week) ULLPT on the 30 m sprint speed of young female and young male soccer players, and the findings demonstrated a similar decrease in 30 m sprint times. Thus, based on the findings, it seems clear and strongly supported that ULLPT helps athletes improve linear sprint speed.

### 4.4. Effect of ULLPT on Agility

This meta-analysis revealed a significant large-sized (ES = 1.31) effect of ULLPT on change of direction speed in athletes compared with active controls. The findings of three reviews with varied inclusion criteria than the current study were similar to ours, confirming an increase in agility [95,96,97]. Indeed, gains in the change of direction speed after plyometric training might be associated with (1) increased eccentric strength of the quadriceps muscles; (2) more efficient braking capacity; (3) enhanced muscle force output and movement efficiency (i.e., lowers ground reaction times) [98,99,100]. Besides, plyometric training may improve mental preparation before high-intensity exercise [101], allowing athletes to perform better in speed-changing activities. In the present review, five papers assessed agility and reported statistically significant improvement in outcomes among athletes from different sports backgrounds (i.e., soccer, swimming, and handball) [32,54,58,60,63]. Karadenizli [60] stated that the ten-week ULLPT program included horizontal and vertical jump and sprint drills, which positively affected the agility performance of young female handball players. Agility improvements in soccer players were shown to be gender-neutral after six weeks of ULLPT by Ramrez-Campillo et al. [58]. Meanwhile, soccer players engaged in plyometric training to create quick ground contact periods and a high reactive strength index, which predicts agility performance [102]. Moreover, reports indicate that explosive strength, muscle coordination, balance, and flexibility impact on agility [98,103]. Additionally, in the literature, it has been well-identified that plyometric training is a time-efficient, practical and straightforward method for improving agility in other sports players [67,104,105]. For example, Asadi [104] discovered that a six-week plyometric training program improves agilityperformance in young male basketball players. According to Pienaar and Coetzee [67], a four week program that combined plyometric exercises and regular rugby practice led to significant increases in agility among university-level rugby players. Fernandez-Fernandez et al. [105] reported that the plyometric training group improved significantly in this parameter following the training intervention. As a result, given the findings from previous plyometric training research that consistently showed favorable improvements in agility among athletes, the positive effects of ULLPT are expected.

### 4.5. Effect of ULLPT on Flexibility

Our findings revealed a significant, small-sized (ES = 0.48) effect of ULLPT on flexibility. Previous research has validated this finding. Studies that were included in a systematic review by Silva et al. [106], showed that plyometric training appears to increase flexibility in volleyball players. da Silva et al. [107] reported evidence that plyometric training may be effective in increasing flexibility in female Futsal athletes. Sáez de Villarreal et al. [108] state that seven weeks of specific plyometric training produced improvements in the flexibility of high-school basketball players. Of note, five of the papers in our review reported on flexibility performance, however, three of them found no substantial improvements in this physical factor [54,60,61]. Karadenizli [60] discovered that the ULLPT group achieved considerable increases on the flexibility test within the group, whereas no difference in flexibility performance was identified between the two groups (ULLPT vs. conventional training) among female handball players. Similarly, Sadeghi et al. [54] found that following a six-week ULLPT program, the results of flexibility in male swimmers revealed no significant changes. It was also shown that there was no significant change in the flexibility performance of male judokas after the ten weeks of ULLPT [61]. Karadenizli [60] speculated that the somatometrics characteristics of participants might impact flexibility performance. Afyon [109] explains that players lack proper flexibility exercise training throughout the training process, and that the cool-down phase after training affected the improvement of flexibility values. In turn, O’Sullivan et al. [110] found evidence that a training strategy based on eccentric contractions can improve lower limb flexibility. Thus, the flexibility increases may be explained by ULLPT’s significant eccentric component in its repetitive high-intensity stretch-shortening cycles [111], which promotes many peripheral adaptations, such as muscular hypertrophy [108]. Furthermore, the increase in flexibility following plyometric training may be connected to dynamic stretching [112], which is included at the end of each ULLPT session. Dynamic stretching elevates muscle temperature, activates the neurological system, increases muscle elasticity, and enhances intramuscular coordination [112,113]. In general, low levels of flexibility may increase the risk of injury, and recent studies have verified the preventative advantages of flexibility exercises when combined with conditioning and strength training regimens [114,115]. Faigenbaum et al. [116] emphasized this idea further by mentioning that stretching should be integrated into a plyometric training plan if increased flexibility is demanded. This should be considered in future experimental research. Thus, although our finding provides evidence that ULLPT may help athletes to gain flexibility advantages, further research is necessary to elucidate the effectiveness of this parameter.

### 4.6. Effect of ULLPT on Balance

Although our results revealed ULLPT-related improvements in measures of static and dynamic balance (ES = 0.10–0.44, trivial–small) compared to active controls, the balance performance increase was not significant. Research conducted by Ramachandran et al. [34] revealed that plyometric jump training had significant small effects on static and dynamic balance in healthy participants (ES = 0.49–0.50). Other individual studies have found no significant gains in balance from solely plyometric jump training interventions [117,118]. Balance is not only required for completing routine activities and avoiding falls in non-athletic groups, but it is also essential for successfully executing sport-specific skills in sports people [119]. Balance impairments were linked to a higher risk of injuries (e.g., ligament strains, ankle sprains) among participants from a variety of sports [120]. Indeed, the enhancements in balance can be linked to lower-limb muscle agonist–antagonist co-activation [121] or alterations in proprioception and neuromuscular control [122]. Interestingly, four studies in our review measured both static and dynamic balance performance [32,53,60,63]; three of them involved female handball players, and one of them employed male basketball players. Dynamic balance seems to be essential and useful in basic handball skill motions such as faking, throwing, and dribbling [60]. However, Hammami et al. [63] did not report a significant improvement in the dynamic balance test, but did observe a significant improvement in the static balance of female handball players after ten weeks of ULLPT. Additionally, Hammami et al. [32] discovered no effects on static and dynamic balance following nine weeks of ULLPT in young female handball players, and Karadenizli et al. [60] revealed that a ten-week ULLPT program improved static balance-unipedal (left) scores in young female handball players. Conversely, there are no considerable effects in dynamic balance-bipedal and static balance-unipedal (right) performances. Hammami et al. [32] explain that the lack of improvement in balance was partly due to the absence of the design of specialized balance exercises during training and partly because of many measurement errors. Another study conducted by Sharma and Multani [53] suggested that ULLPT is more efficient than normal basketball practice in enhancing the static balance of male basketball players. Previously, several studies indicate that using a combinational plan (plyometric + balance) could be more effective in improving balance. For example, Bouteraa et al. [123] reported a large increase in the stork balance and dynamic performances of female adolescent basketball players after 8 weeks of combining plyometrics with balance training. Lu et al. [124] concluded that six weeks of combined training can induce greater benefits for dynamic balance among elite male badminton athletes. Therefore, more studies evaluating the effect of ULLPT on balance are needed to draw more solid conclusions.

### 4.7. Additional Analysis

Our findings showed that age, sex, and training program variables did not influence the effect of ULLPT on athletes’ lower body muscle power (CMJ, CMJa), upper body muscle strength, linear sprint speed (30 m), agility, and flexibility performance. Previous meta-analyses of plyometric training found no significant subgroup differences or relationships between participants’ age and sex [22,33]. Moreover, the current meta-analysis includes one trial [58] that evaluated the effects of ULLPT on females and males, with both genders showing similar gains in fitness parameters, including CMJ height, agility, and sprint; the authors elaborated that gender should not be considered as a particular concern when implementing plyometric training in adult soccer players, at least when the objective is increasing specific physical performance [58]. However, other previous works have reported that the athlete’s age and sex may influence the adaptive response to plyometric training in contrast to our findings [49,50,66]. Typically, designing an optimal plyometric training program is related to training load (volume and intensity) and the frequency of training [104]. A significant positive relationship has been reported between the program duration, the number of sessions, and the plyometric training effect by Moran [50] and de Villarreal et al. [66]. The research team led by Ramirez-Campillo et al. [74] suggested that practitioners should attempt to periodize plyometric training for athletes by adjusting both volume and intensity to achieve meaningful changes. Yet, few controlled trials in the literature [125,126] have looked at the potential effects of different plyometric training intensities on fitness measures in athletes and/or physically active populations. Additionally, in this review, we found no significant associations between these variables and physical fitness component gains. The preliminary studies on some moderators may have limited our exploration of moderating factors on physical fitness adaptations after ULLPT. Consequently, we are unable to confidently advise athletes on the optimal training factors for ULLPT to improve their physical fitness. Further investigation of the moderating factors of ULLPT is required in the future.

## 5. Limitations

Some potential limitations should be mentioned and discussed. First, only 15 studies were included in the present review; this may limit the analysis of larger data sets. For example, a meta-analysis for endurance (e.g., Yo-Yo test, 20 m shuttle-run test) could not be carried out because less than three studies evaluated measures of this variable. Second, additional analyses regarding ULLPT frequency, length, total sessions, time, and intensity were not always possible because in some cases there were fewer than three papers available for at least one of the moderators. Therefore, we cannot firmly suggest ideal training variables to improve athletes’ physical fitness. Third, the median split strategy might potentially lead to residual confounding and a reduction in statistical power when dichotomizing continuous data (e.g., 15 years vs. >15 years) [127]. Fourth, in this meta-analysis, the training protocols were not wholly consistent in each study, which may be one of the main reasons for the high level of heterogeneity (e.g., upper body muscle strength, agility). Finally, some studies did not provide a complete description of the training program; for example, eight papers (out of fifteen) did not specify the surface (e.g., grass) on which ULLPT was used. The choice of the training surface can affect injuries, as well as change the effects of plyometric training [128].

## 6. Practical Applications

ULLPT has been proven to have training advantages for both upper and lower body physical performance of athletes; indeed, our data indicates not only increased flexibility, upper body muscle strength and power, but also substantial gains in lower body muscle power, sprint speed, and agility. The improvements that we found should arouse the interests of coaches, other researchers, and physical education experts since many sports rely greatly on these physical fitness components, all of which were boosted by the plyometric training regimen for both the upper and the lower limbs. Therefore, we recommend ULLPT as an effective training option to improve athletes’ several key physical fitness, as mentioned above. In terms of the characteristics of optimal ULLPT regimens, a stimulus of 2–3 sessions per week, lasting 20–45 min each, for 4–10 weeks seems to be adequate in increasing athletes’ physical fitness. Moreover, if the training program is properly designed and professionally supervised, it will not lead to an additional overload on the athlete’s musculoskeletal system, as evidenced by no paper reporting injuries throughout the program implementation; thus, ULLPT may be regarded as an integral aspect of neuromuscular training sessions focusing on upper and lower extremity injury prevention. Furthermore, based on our findings, sole use of ULLPT may not be recommended for improving balance in athletes, but the reasons remain unclear; more ULLPT research on this factor is needed.

## 7. Conclusions

This systematic review with meta-analysis summarizes existing research on the favorable benefits of the ULLPT program on several key physical fitness in athletes, including tennis players, handball players, judokas, gymnasts, volleyball players, basketball players, soccer players, and swimmers. According to this review, ULLPT induces distinct neuro-muscular adaptations in the upper and lower body musculature and is an efficient method for enhancing athletes’ physical fitness (i.e., upper and lower body muscle power, linear sprint speed, upper and lower body muscle strength, agility, and flexibility). An absence of evidence for significant improvements of ULLPT on balance was found. Due to a small number of studies (less than three per moderator), more research is needed in this area to determine whether gender, age, and training factors following ULLPT have potentially moderating effects on physical fitness components among athletes. Moreover, we would encourage further research to explore the effects of ULLPT on more physical fitness parameters (e.g., endurance, coordination, reaction time, body composition), in order to have a better understanding of the advantages of this kind of training on the overall physical fitness of athletes.

## Figures and Tables

**Figure 1 ijerph-20-00482-f001:**
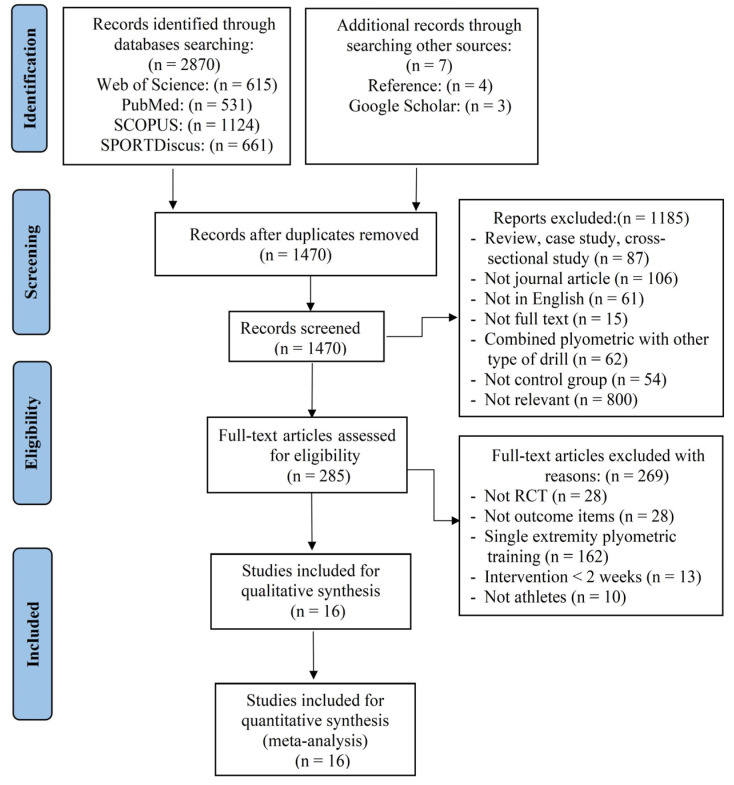
Flow diagram of the search process.

**Figure 2 ijerph-20-00482-f002:**
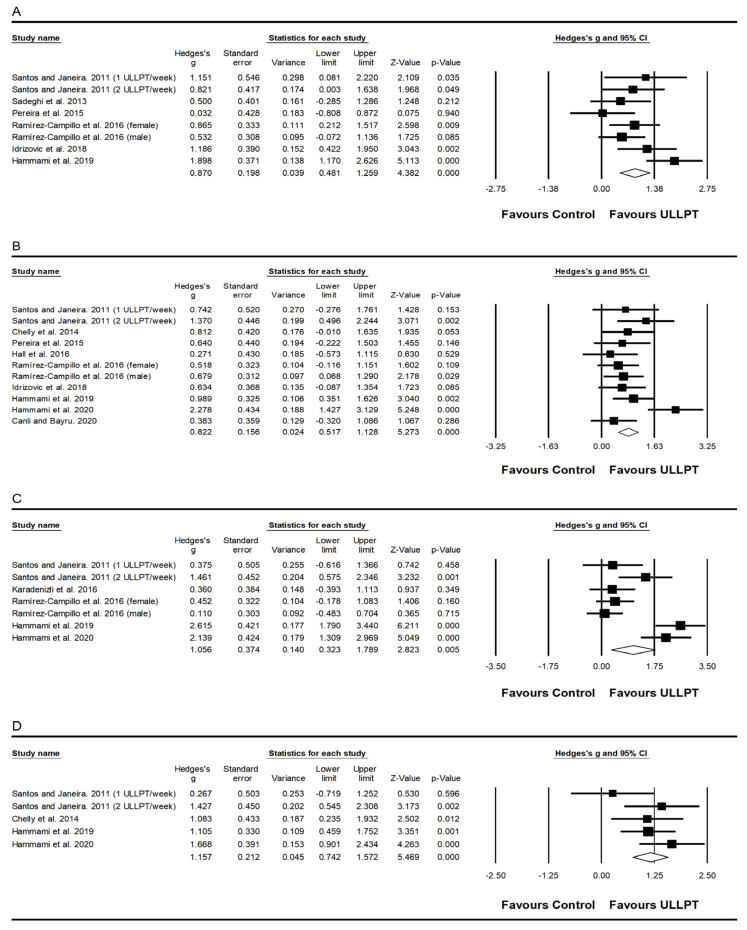
Forest plot of changes in MBT performance (**A**) [32,52,54,57,58,62], CMJ performance (**B**) [31,32,52,56,57,58,59,62,63], CMJa performance (**C**) [32,51,52,58,60], and squat jump performance (**D**) [32,52,56,63], in athletes participating in combined upper and lower limb plyometric training (ULLPT) compared to controls. Values shown are effect sizes (Hedges’ g) with 95% confidence intervals (CI). The size of the plotted squares reflects the statistical weight of the study.

**Figure 3 ijerph-20-00482-f003:**
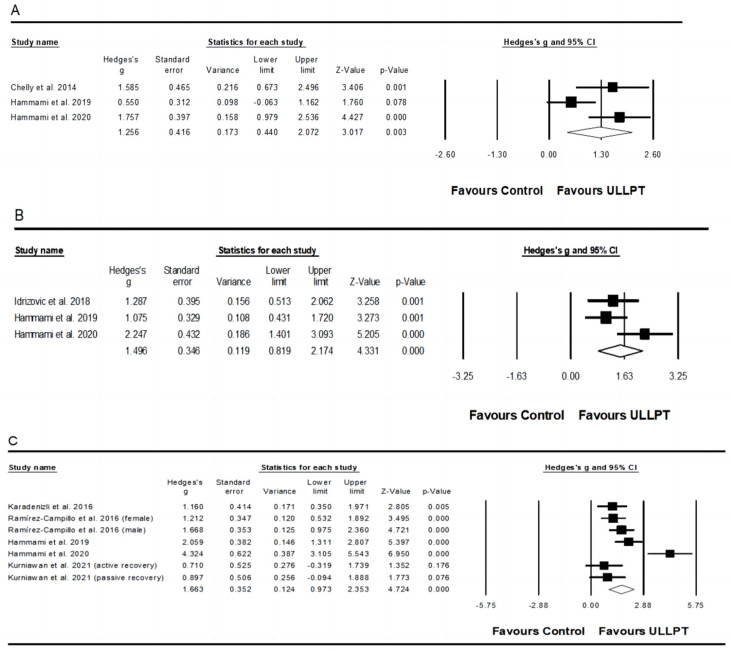
Forest plot of changes in 5 m sprint performance (**A**) [32,56,63], 20 m sprint performance (**B**) [32,62,63], and 30 m sprint performance (**C**) [32,58,60,63,64], in athletes participating in combined upper and lower limb plyometric training (ULLPT) compared to controls. Values shown are effect sizes (Hedges’s g) with 95% confidence intervals (CI). The size of the plotted squares reflects the statistical weight of the study.

**Figure 4 ijerph-20-00482-f004:**
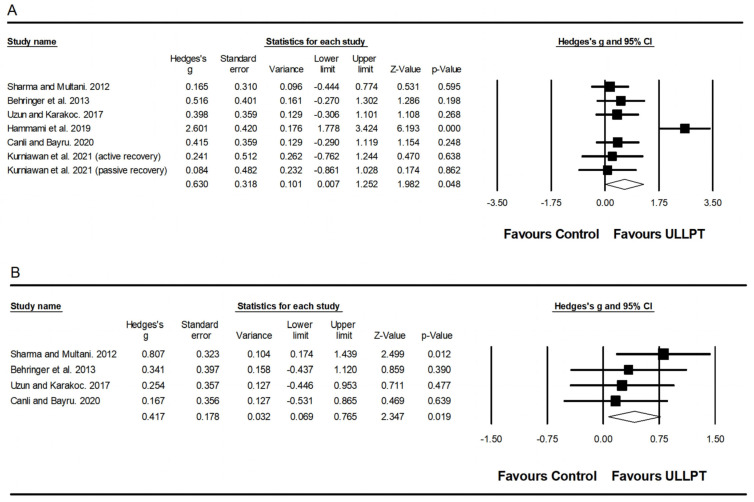
Forest plot of changes in upper body strength performance (**A**) [31,32,53,55,61,64] and lower body strength performance (**B**) [31,53,55,61] in athletes participating in combined upper and lower limb plyometric training (ULLPT) compared to controls. Values shown are effect sizes (Hedges’s g) with 95% confidence intervals (CI). The size of the plotted squares reflects the statistical weight of the study.

**Figure 5 ijerph-20-00482-f005:**
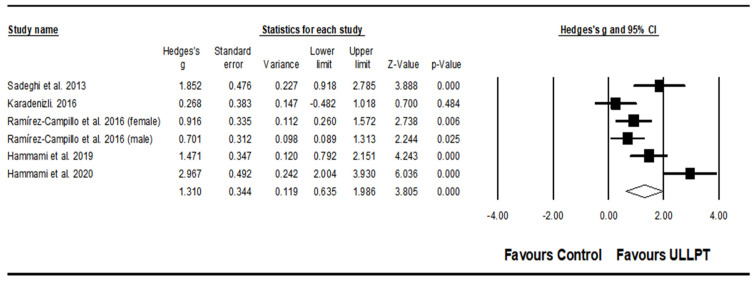
Forest plot of changes in agility performance [32,54,58,60,63] in athletes participating in combined upper and lower limb plyometric training (ULLPT) compared to controls. Values shown are effect sizes (Hedges’s g) with 95% confidence intervals (CI). The size of the plotted squares reflects the statistical weight of the study.

**Figure 6 ijerph-20-00482-f006:**
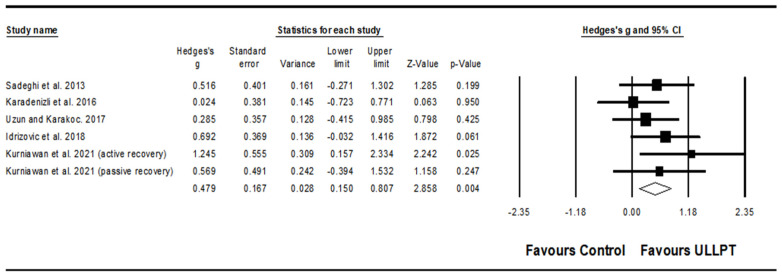
Forest plot of changes in flexibility performance [54,60,61,62,64] in athletes participating in combined upper and lower limb plyometric training (ULLPT) compared to controls. Values shown are effect sizes (Hedges’s g) with 95% confidence intervals (CI). The size of the plotted squares reflects the statistical weight of the study.

**Figure 7 ijerph-20-00482-f007:**
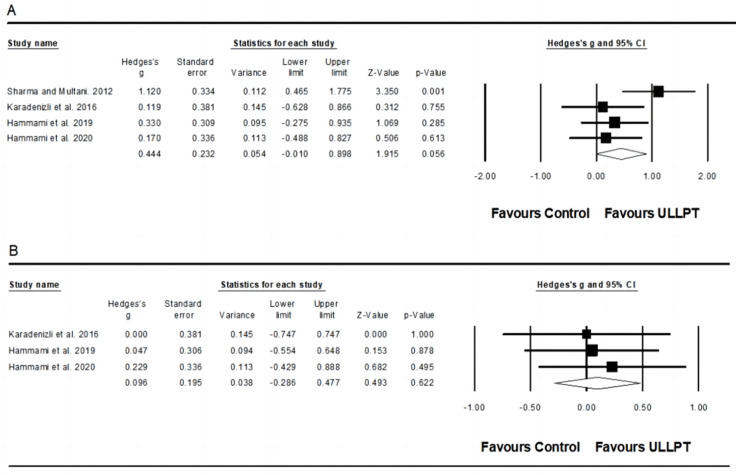
Forest plot of changes in static balance performance (**A**) [32,53,60,63] and dynamic balance performance (**B**) [32,60,64] in athletes participating in combined upper and lower limb plyometric training (ULLPT) compared to controls. Values shown are effect sizes (Hedges’s g) with 95% confidence intervals (CI). The size of the plotted squares reflects the statistical weight of the study.

**Table 1 ijerph-20-00482-t001:** Selection criteria used in the meta-analysis.

Category	Inclusion Criteria	Exclusion Criteria
Population	Athletes, with no restrictions on their sport activity, sex, or age.	Athletes with health problems (e.g., injuries, recent surgery).
Intervention	A ULLPT program, defined as combined upper-limb plyometrics (medicine ball exercises, push-ups, and chess press) and lower-limb plyometrics (unilateral or bilateral bounds, jumps, hops, and/or skips) (Not less than 2 weeks).	Plyometric training combined with other forms of training (e.g., resistance training, sprint training) or single limb plyometrics.
Comparator	Active control group.	Absence of active control group
Outcome	At least one measure related to physical fitness (e.g., power, sprint, strength) before and after the training intervention.	Lack of baseline and/or follow-up data.
Study design	Randomized Controlled Trials.	Non-Randomized Controlled Trials.

Note: ULLPT, combined upper and lower limb plyometric training.

**Table 2 ijerph-20-00482-t002:** Physiotherapy Evidence Database (PEDro) scale ratings.

PEDro Scale Items *	No. 1	No. 2	No. 3	No. 4	No. 5	No. 6	No. 7	No. 8	No. 9	No. 10	No. 11	Total (from a Possible Maximal of 10)
Santos and Janeira, 2011 [52]	1	1	0	1	0	0	0	1	1	1	1	6
Sharma and Multani, 2012 [53]	0	1	0	1	0	0	0	0	0	1	1	4
Sadeghi et al., 2013 [54]	0	1	0	1	0	0	0	1	0	0	1	4
Behringer et al., 2013 [55]	1	1	1	1	0	0	0	1	1	1	1	7
Chelly et al., 2014 [56]	1	1	0	1	0	0	0	1	1	1	1	6
Pereira et al., 2015 [57]	0	1	0	1	0	0	0	1	0	1	1	5
Ramírez-Campillo et al., 2016 [58]	0	1	1	1	1	0	0	1	0	1	1	7
Hall et al., 2016 [59]	1	1	0	1	0	0	0	1	1	1	1	6
Karadenizli, 2016 [60]	1	1	0	1	0	0	0	1	1	1	1	6
Uzun and Karakoc, 2017 [61]	0	1	0	1	0	0	0	1	1	1	1	6
Idrizovic et al., 2018 [62]	1	1	0	1	0	0	0	1	1	1	1	6
Nowakowska et al., 2017 [51]	0	1	0	1	0	0	0	0	0	0	1	3
Hammami et al., 2019 [32]	1	1	1	1	0	0	0	1	1	1	1	7
Hammami et al., 2020 [63]	1	1	0	1	0	0	0	1	1	1	1	6
Canlı and Bayru, 2020 [31]	1	1	0	1	0	0	0	1	1	0	1	5
Kurniawan et al., 2021 [64]	0	1	0	1	0	0	0	1	1	1	1	6

Note: * A detailed explanation for each PEDro scale item can be accessed at https://www.pedro.org.au/english/downloads/pedro-scale (accessed on 1 October 2022).

**Table 3 ijerph-20-00482-t003:** Characteristics of included study participants in experimental groups.

References	Athletes	N	Gender	Age	Body Mass	Height	SPT	Fitness
Santos and Janeira, 2011 (2ULLPT/week) [52]	Basketball players	14	M	15.0	62.6	172	No	NR
Santos and Janeira, 2011 (1ULLPT/week) [52]		7						
Sharma and Multani, 2012 [53]	Basketball players	20	M	12–20	NR	NR	NR	NR
Sadeghi et al., 2013 [54]	Swimmer	12	F	12.23	41.92	144.53	NR	NR
Behringer et al., 2013 [55]	Tennis players	10	M	15.5	65.2	177	NR	Normal-moderate
Chelly et al., 2014 [56]	Handball players	12	M	17.1	80.1	181	NR	Moderate
Pereira et al., 2015 [57]	Volleyball players	10	F	14.0	52.0	160	No	Moderate
Ramírez-Campillo et al., 2016 [58]	Soccer players	19/21	F/M	22.4/20.4	60.7/68.4	161/171	No	Moderate
Hall et al., 2016 [59]	Gymnasts	10	F	12.5	40.5	146	NR	Moderate
Karadenizli, 2016 [60]	Handball players	14	F	15.64	54.41	161	NR	Moderate
Uzun and Karakoc, 2017 [61]	Judokas	15	M	21.40	71.26	176	NR	NR
Idrizovic et al., 2018 [62]	Volleyball players	13	F	16.6	59.4	173	No	High
Hammami et al., 2019 [32]	Handball players	21	F	13.5	42.6	142	NR	Moderate
Hammami et al., 2020 [63]	Handball players	17	F	15.8	64.2	166	Yes	High
Canlı and Bayru, 2020 [31]	Basketball players	15	M	14.7	67.6	174	NR	Moderate
Kurniawan et al., 2021 (active recovery) [64]	Judokas	11	M	21.8	71.1	170	NR	Normal
Kurniawan et al., 2021 (passive recovery) [64]		11		21.7	63.8	171		

F, female; M, male; N, number of participants; NR, not reported; Fitness: high, for professional/elite athletes with regular enrolment in national and/or international competitions, highly trained participants with >10 training hours per week or >6 training sessions per week and scheduled official and friendly competitions. Moderate, for non-elite/professional athletes, with regular attendance in regional and/or national competitions, between 5 and 9.9 training hours per week or 3–5 training sessions per week and scheduled official and friendly competitions. Normal, for recreational athletes with <5 training hours per week with sporadic competitions participation, and physically.

**Table 4 ijerph-20-00482-t004:** Characteristics of ULLPT interventions.

References	Freq	Dur	Time	Int	Training Protocol		RBSE (s)	RBR (s)	RBTS (h)	Surf	TP	PO	Rep
					Upper Limb	Lower Limb							
Santos and Janeira, 2011 [52] (2 ULLPT/week)	2	10	NR	NR	MB exercises (chest/pullover pass, power drop, squat toss, seated backward throw, backward throw) 2–4 sets × 6–10 reps	Rim/squat/tuck/side/box-to-box jump, depth jump with or without 180-degree turn, single- arm alternate-leg bound, hurdle/two-foot ankle hop, zigzag drill, alternate leg push-off, lateral jump over cone, cone hops with COD sprint 2–4 sets × 5–15 reps	60–240	15–90	NR	NR	IS	C	No
Santos and Janeira, 2011 [52] (1 ULLPT/week)	1	16	NR	NR	MB exercises (pullover pass, power drop) 4 sets × 10 reps	Depth jump 180-degree turn, hurdle hops, cone hops with COD sprint, multiple box-to-box jumps 4 sets × 6–10 reps	60–240	15–90	144	NR	IS	V.T	NO
Sharma and Multani, 2012 [53]	3	4	NR	Low, moderate and high *	MB exercises (back toss, overhead /side/squat/start-throw, over back toss), push-ups 2 sets × 5 reps	Squat/tuck/depth jump, jump to box, bounding with rings, single leg lateral hops 2 sets × 40 ground contacts	NR	NR	NR	Polo	NR	V.I.T	No
Sadeghi et al., 2013 [54]	2	6	NR	Low, moderate and high *	No detailed description	No detailed description	NR	NR	NR	NR	NR	V.I	No
Behringer et al., 2013 [55]	2	10	45 min	NR	Push-ups with and without clapping hands, MB chest pass, two-hand overhead throw with and without upper body rotation 3–4 sets × 10–15 reps	rope skipping/lateral barrier hop (single-and double-leg), box hopping (clock- and counter-clockwise; single- and double-leg), cycled split squat jump, countermovement jump, countermovement jump to box, 3–4 sets × 10–15 reps	20–60	0–1	55–78	NR	NR	C	Yes
Chelly et al., 2014 [56]	2	8	30 min	Max	Dynamic push-up 3–4 sets × 10–12 reps	Hurdle/dop jumps 4–10 sets × 10 reps	NR	NR	48	NR	IS	C	Yes
Pereira et al., 2015 [57]	2	8	20 min	Max	Unilateral MBT, MBT 2 sets × 6 reps	Bilateral jump (with or without bending knees), unilateral jump (with the dominant leg on the floor) 3–5 sets × 10–25 reps	120	NR	48	NR	IS	V.I	No
Ramírez-Campillo et al., 2016 [58]	2	6	30 min	Max	MBT 3 sets × 8 reps	Cyclic and acyclic horizontal and vertical jumps, with left, right and both legs 2 sets × 5 reps	60	15	72	Grass	IS	V	Yes
Hall et al., 2016 [59]	2	6	40 min	Max	Chest pass, single-arm/sit-up MBT, inverted clap push-ups, push-up on and off raised surface 2–4 sets × 1–5 reps	Tuck/split/squat jump, jump over barrier, (15/30 cm), multiple box-to-box jumps, single leg bounding, jump to/from Box (30 cm), standing long jump, handstand/shoulder shrug hops, alternate leg push- off, bounce to handstands against wall 1–4 sets × 1–6 reps	60	NR	NR	Concrete	NR	C	No
Karadenizli et al., 2016 [60]	2	10	NR	Max	Overhead passing with MB, sit-up, overhead throwing with handball ball 2–4 sets × 10 reps	Forward/side to side skipping over cone with or without MB, side to side ankle hops/skipping, slalom running and sprint, double leg front jump over hurdle, standing vertical-jump and reach, double/single leg forward- jump over hurdle, horizontal jump and sprint, Single leg diagonal/forward/lateral-jump 2–4 sets × 3–15 reps	60–180	NR	48–120	NR	IS	V.T	No
Uzun and Karakoc, 2017 [61]	3	10	20 min	Max	No detailed description	No detailed description	NR	NR	NR	NR	NR	V.I	No
Idrizovic et al., 2018 [62]	2	12	20–30 min	Max	MB press, MB alternating throw, chest pass, push-ups, jumping spider (from knees), overarm throws 2–4 sets × 2–5 reps	Stiff knee leg hops, vertical/tuck jumps, lateral/diagonal jumps, broad jumps, obstacle/box drop 3–5 sets × 1–5 reps	120–300	NR	168	Wood	PS	V.T.I	No
Hammami et al., 2019 [32]	2	9	NR	Low, moderate and high *	Dynamic push-up 10 sets × 6–8 reps	hurdle jump, stretched leg jump, lateral hurdle jump 10 sets × 6–8 reps	90	NR	48	NR	NR	V.T.I	Yes
Hammami et al., 2020 [63]	2	10	NR	Max	Push-up 10 sets × 6 reps	Horizontal/stretched leg/hurdle jump, lateral hurdle jump 2–3 sets × 6 reps	30–60	NA	48	NR	IS	V.T.I	Yes
Canlı and Bayru, 2020 [31]	2	9	30–35 min	Max	Shoulder/overhead press, MBT, bench press with theraband, push-up, side shuffle with chest press, step-up, ladder with MB, burpee	Box/broad/squat/lateral box/jumps, Jumping lunges, Front to back hurdle hop, skater hoop	120–180	25–30	72	NR	IS	V.T	No
Kurniawan et al., 2021 [64]	3	8	NR	NR	No detailed description	No detailed description	NR	NR	NR	NR	NR	NR	No

Int, intensity; Max, maximal, involving either maximal effort to achieve maximal height, distance, reactive strength index, velocity (time contact or fast stretch-shortening cycle), or another marker of intensity. For the studies marked with an *, the intensity was reported only qualitatively; PO, progressive overload, in the form of either volume (i.e., V), intensity (i.e., I), type of drill (i.e., T), or a combination of these; Rep, replacement of a portion of the habitual training drills with plyometric jump training drills; RBR, rest between repetitions (seconds); RBSE, rest between sets and/or exercises (seconds); RBTS, rest between training sessions (hours); Surf, surface type; Freq, frequency of PJT (days/week); Dur weeks of training; TP, training period. Time, the time of ULLPT session itself (main part). MB, medicine ball; MBT, medicine ball throwing.

## Data Availability

The datasets generated and analyzed for this study can be requested by correspondence authors.

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
