# Peer review of "Effects of Combined Upper and Lower Limb Plyometric Training Interventions on Physical Fitness in Athletes: A Systematic Review with Meta-Analysis"

_ijerph, 2022, doi:10.3390/ijerph20010482_

Round 1

Reviewer 1 Report

In general, the current study adopted a comprehensive Meta-Analysis software (v3 Biostat, USA) to examine the effects of combined upper and lower limb plyometric training (ULLPT) on physical fitness attributes in athletes, via a systematic literature-based, searched 15 studies whose quality is verified from moderate-to-high including total 523 participants & aged 12-22.4 years for the analyses. Although the study properly included baseline-controlled studies with follow-up measures to examine one measure of physical fitness indices in athletes for the ULLPT effects, the figures 2~7 of this 3.5-section meta-analysis results are unreadable and should be revised accordingly.

Minor comments

Line 110~111 “N NNote: ULLPT,…” not clear?

Line 156~158, “Based on the method described in … studies [22,35]. If three or more relatively homogeneous studies were submitted with baseline and follow-up data for the same parameter, studies were meta-analytically pooled.” The sentence could have been shortened for clarity.

Line 199, figure 1’s Screening: Records after duplicates (n=1407) removed (n-1407). This n is questionable.

Line 231~233. The abbreviations, RBR; RBSE, BTS, Surf, surface type do not well correspond to Table 4’s acronyms listed.

Line 296, the section number of the Effect of ULLPT on Balance is not correct, including its following section number as well.

Author Response

Point 1: The figures 2~7 of this 3.5-section meta-analysis results are unreadable and should be revised accordingly.

Response 1: We apologize for the figure’s problems in the original manuscript. And thank you for pointing this out, we have revised and improved all figures in the resubmitted manuscript. We did not track the changes, but we have removed the old figures. To avoid the figures’ problem, we have provided the original high-resolution figures in the supplementary file. See “Forest Plots”.

Point 2: Line 110~111 “N NNote: ULLPT,…” not clear?

Response 2: Thank you for your careful reading. We have corrected this mistake. Line 115.

Point 3: Line 156~158, “Based on the method described in … studies [22,35]. If three or more relatively homogeneous studies were submitted with baseline and follow-up data for the same parameter, studies were meta-analytically pooled.” The sentence could have been shortened for clarity.

Response 3: Thank you for your valuable suggestion. We have rewritten this sentence in the resubmitted manuscript. Line 166-167.

Point 4: Line 199, figure 1’s Screening: Records after duplicates (n=1407) removed (n-1407). This n is questionable.

Response 4: We sincerely appreciate your comments. The number of duplicates removed is 1407. After removing duplicates, 1470 research papers remained. Perhaps these two numbers together may confuse the reader, so we have modified this part to show only the number after removing duplicates. See Figure 1 (we already removed the old one).

Point 5: Line 231~233. The abbreviations, RBR; RBSE, BTS, Surf, surface type do not well correspond to Table 4’s acronyms listed.

Response 5: Thank you for your careful reviewing. We are sorry for our carelessness; we have revised the abbreviations. See Table 4 and lines 256-257.

Point 6: Line 296, the section number of the Effect of ULLPT on Balance is not correct, including its following section number as well.

Response 6: We were really sorry for our careless mistakes. In our resubmitted manuscript, we have revised it. Line 323.

Reviewer 2 Report

Interesting and valuable meta-analysis of a wide field of research. The method is well designed and the inclusion/exclusion criteria as well as the limitations are well defined.

Some English language check is required. Please rephrase lines 24-25 and 32-34. The abstract would be more clear if titles were added for clarity.

Figures 2,3,4,5,6,7 were illegible and should be corrected.

The authors have successfully performed a systematic review and a meta-analysis on the literature and results of Plyometric Training Interventions on Physical Fitness in Athletes. The topic is original and presents scientific interest since there is a vast amount of literature on the subject that should be re-assesed. The conclusions are consistent with the evidence and arguments presented and the issue posed is addressed. No issues were detected with the references and there is a good description of the exclusion and inclusion criteria. All in all, the method designed is appropriate to address the question posed. However, the figures should be better presented since they are illegible. There is a need of rephrasing in lines 24-25 and 32-34. The abstract would be benefitted from the addition of titles that would add clarity. Furthermore there is a need for minor check of the use of the language. I hope that this will suffice. Thank you again for the opportunity to review this manuscript. 

Author Response

Point 1: Some English language check is required. Please rephrase lines 24-25 and 32-34. The abstract would be more clear if titles were added for clarity.

Response 1: Thank you for your valuable suggestions. We have rewritten these sentences in the resubmitted manuscript. Line 24-25, line 33-35. And we have added the titles in the abstract part. See the abstract part.

Point 2: Figures 2,3,4,5,6,7 were illegible and should be corrected.

Response 2: We apologize for the figure’s problems in the original manuscript. And thank you for pointing this out, we have revised and improved all figures in the resubmitted manuscript. We did not track the changes, but we have removed the old figures. To avoid the figures’ problem, we have provided the original high-resolution figures in the supplementary file. See “Forest Plots”.

Point 3: The figures should be better presented since they are illegible. There is a need of rephrasing in lines 24-25 and 32-34.

Response 3: We sincerely appreciate the valuable comments. We have rephrased these sentences (lines 24-25 and 33-34) in the resubmitted manuscript. Line 24-25, line 34-36.

Point 4: The abstract would be benefitted from the addition of titles that would add clarity. Furthermore there is a need for minor check of the use of the language. I hope that this will suffice.

Response 4: Thank you for your suggestion. We carefully checked the entire manuscript for typographic, grammatical, and formatting errors in the resubmitted paper. Such as line 89, line 174, line 182. We have added the titles in the abstract part. See the abstract part.

Reviewer 3 Report

Authors have provided systematic review on the topic with details of procedures written in the article. 

My suggestions:

1. Replace "change of direction speed (CODS)" by "agility". Agility is a more commonly used term in sport field than CODS for describing motions involving change of direction. 

2. The p value in the article should be written as italic "p"

3. In sentence 104, Table S1 has been written in the sentence but I couldn't find it.

4. The information in figure 2-7 is so unclear that I cannot read it.

This article has a clear aim to reveal the benefits of ULLPT on different physical fitness parameters (power, strength, change of direction speed (agility), flexibilty and balance) for athletes. It followed the PRISMA guidelines to conduct the meta-analysis systemtaically. 15 studies were selected for further analysis finally which included 523 participants. Most of the essential elements of meta-analysis, such as details of inclusion and exclusion criteria, quality of edvidence, data extraction and results presentation were clearly stated in  the article. However, the figures of forest plot (figure 2-7) were unclear and affected the readability of the article.  

Minor comments: 

     All figures should be reviewed to ensure that it can be seen by readers clearly.  
    Line 139, "change of direction speed" is less commonly used term to mention about physical fitness is sport field. It is recommended to change it into "agility". 
    Line 105, "research period" of the articles selcted is recommended to include  in section 2.2 eligibility criteria 
    In the abstract, authors mentioned that training program variables is included in the study as modulator. However, in line 320, only the lenght of the training program is included. More variables of the trianing program, such as frequency, intensity, time should be included in the analysis.  
    The focus of this study is about physical fitness. However, in the second paragraph of 6.1, authors discussed a lot about the "jump abilities" which should not be considered as physical fitness as there several fitness components consitutes "jump abilities".Authors are suggested to provide details explanation of this part regarding to each fitness components  concerned.  
    Line 292, the forest plot should be about flexibility but not anaerobic power performance.

Author Response

Point 1: Replace "change of direction speed (CODS)" by "agility". Agility is a more commonly used term in sport field than CODS for describing motions involving change of direction.

Response 1: Thank you for your valuable suggestion. We have replaced "change of direction speed (CODS)" by "agility" in our resubmitted manuscript. Line 150, 306, 307, line 486 “ Effect of ULLPT on Agility” section.

Point 2: The p value in the article should be written as italic "p"

Response 2: We sincerely appreciate the valuable comments, we have checked the whole paper’s p value and written as italic "p". And here we did not list the changes, but tracked the changes in the revised manuscript. 

Point 3: In sentence 104, Table S1 has been written in the sentence but I couldn't find it.

Response 3: Thank you for your careful reviewing. Table S1 is one of the Supplementary files (search strategy), we submitted together with the manuscript in the journal’s system and we mentioned in lines 693,694. In order to make it clearer to the reader, we have motioned “Supplementary Online, Table S1” in our resubmitted paper. Line 261, line 108.

Point 4: The information in figure 2-7 is so unclear that I cannot read it.

Response 4: We apologize for the figure’s problems in the original manuscript. And thank you for pointing this out, we have revised and improved all figures in the resubmitted manuscript. We did not track the changes, but we have removed the old figures. To avoid the figures’ problem, we have provided the original high-resolution figures in the supplementary file. See “Forest Plots”.

Point 5: The figures of forest plot (figure 2-7) were unclear and affected the readability of the article.  All figures should be reviewed to ensure that it can be seen by readers clearly.  

Response 5: We sincerely appreciate your comments. We apologize for the figure’s problems in the original manuscript. And thank you for pointing this out, we have revised and improved all figures in the resubmitted manuscript. See figures 1-7. To avoid problems with the figures again, we have provided the original high-resolution figures (figures 2-7) in the supplementary file. See “Forest Plots”.

Point 6: Line 139, “change of direction speed”; is less commonly used term to mention about physical fitness is sport field. It is recommended to change it into “agility”;. 

Response 6: Thank you for your valuable suggestion. We have replaced all "change of direction speed (CODS)" by "agility" in our resubmitted manuscript.

Point 7: Line 105, “research period” of the articles selected is recommended to include in section 2.2 eligibility criteria 

Response 7: Thank you for your valuable suggestion. We have added the statement in the revised paper. Line 117-118.

Point 8: In the abstract, authors mentioned that training program variables is included in the study as modulator. However, in line 320, only the lenght of the training program is included. More variables of the training program, such as frequency, intensity, time should be included in the analysis.  

Response 8: Thanks so much for the advice. Based on your comments, we extracted the training time of each study, see table 4, but this set of data are not available for meta-analysis. By carefully reading the included studies, 7 out of 15 studies did not detail reported the time of ULLPT session itself (main part). In line 337-338, we clearly indicate that due to a restricted number of trials (three per moderator), only 28 analyses of moderators were performed (as shown in additional analysis section). Although these variables (i.e., frequency, intensity, time) can be considered as moderating factors, conducting meta-analysis is difficult in the current study due to the inadequate data provided (< 3 studies per moderator).

Nevertheless, we recognize this limitation should be clearly mentioned in the paper, so we added more information in the revised manuscript. Line 196-202, line 631-633.

Point 9: The focus of this study is about physical fitness. However, in the second paragraph of 6.1, authors discussed a lot about the “jump abilities”; which should not be considered as physical fitness as there several fitness components consitutes “jump abilities”; Authors are suggested to provide details explanation of this part regarding to each fitness components concerned.  

Response 9: Thank you for your suggestion. Based on your comment and discuss with our review team, the “jump abilities” should be corrected as “vertical jump height” . Vertical jumps include squat jump (SJ), countermovement jump with (CMJa) or without swing (CMJ) (Markovic, 2007). We have revised and added more information in this part of content. Line 388, line 391-396..

 (Ref: Markovic, G. (2007). Does plyometric training improve vertical jump height? A meta-analytical review. Br. J. Sports Med. 41.349–355. doi:10.1136/bjsm.2007. 035113)

Point 10: Line 292, the forest plot should be about flexibility but not anaerobic power performance.

Response 10: we were really sorry for our careless mistakes, in our resubmitted manuscript, we have revised it. Line 320.

Reviewer 4 Report

This was a thorough review and process put into the study. Would there have been anyway to possibly lessen the exclusion criteria to increase studies in the analysis? Would have liked to see a larger data set without sacrificing quality.  

Many of the tables are difficult to read because of size/shading, recommend enlarging or changing font format.

For each area, there could be more discussion on distinct findings from a single study or small group. This was done, but more could be added to provide additional specifics.

Concluding Thought

This was a strong paper and thorough process to get to these findings. I do not see any significant changes to be made.

Author Response

Point 1: This was a thorough review and process put into the study. Would there have been anyway to possibly lessen the exclusion criteria to increase studies in the analysis? Would have liked to see a larger data set without sacrificing quality.

Response 1: We appreciate your insightful suggestion and agree that lessen the exclusion criteria would be useful to get more data set for analysis; so, we tried to excluded the intervention lasted < 2 weeks (see resubmitted manuscript’s figure 1). We carefully read the 13 papers which contained in “Intervention < 4 weeks” (original paper’s figure 1), however, we did not find additional eligible study, these articles aims to explore the acute effect of plyometric training.

In addition, our aims only to show that the effect of combined upper and lower limb plyometric training intervention on athlete’s physical fitness indices. We believe that the exclusion criteria we have established are reasonable and scientific. For example:

  • We excluded non-randomized controlled trials. Due to meta-analyses of randomized-controlled trials are usually located at the top of evidence hierarchies (El-Rabbany et al., 2017; Maziarz, 2022).

(Ref: Maziarz, M. (2022). Is meta-analysis of RCTs assessing the efficacy of interventions a reliable source of evidence for therapeutic decisions?. Studies in History and Philosophy of Science, 91, 159-167. doi:10.1016/j.shpsa.2021.11.007

El-Rabbany, M., Li, S., Bui, S., Muir, J. M., Bhandari, M., & Azarpazhooh, A. (2017). A quality analysis of systematic reviews in dentistry, part 1: meta-analyses of randomized controlled trials. Journal of Evidence Based Dental Practice, 17(4), 389-398. doi:10.1016/j.jebdp.2017.06.004)

  • In line with previous similar research (Stojanovic et al., 2017), we excluded plyometric training combined with other types of strength training, in order to avoid the effects of combined training.

(Ref: Stojanovic´, E., McMaster, V. R. D. T., and Milanovic´, Z. (2017). Effect of Plyometric Training on Vertical Jump Performance in Female Athletes : A Systematic Review and Meta-Analysis. Sport. Med. 47.975–986. doi:10.1007/s40279-016-0634-6)

  • The inclusion of an active control group was considered essential in order to isolate the effect of plyometric training from the rest of training methods that players commonly conduct in their regular training schedule (Sánchez et al., 2020).

(Ref: Sánchez, M., Sanchez-Sanchez, J., Nakamura, F. Y., Clemente, F. M., Romero-Moraleda, B., and Ramirez-Campillo, R. (2020). Effects of plyometric jump training in female soccer player’s physical fitness: A systematic review with meta-analysis. Int. J. Environ. Res. Public Health. 17.1–23. doi:10.3390/ijerph17238911)

Point 2: Many of the tables are difficult to read because of size/shading, recommend enlarging or changing font format.

Response 2: We were really sorry for the tables problem, in our resubmitted manuscript, we have enlarged Tables 2-4. See Tables 2-4.

Point 3: For each area, there could be more discussion on distinct findings from a single study or small group. This was done, but more could be added to provide additional specifics.

Response 3: Thank you for your valuable suggestion. Based on your comments, we tried our best to improve the manuscript and made some changes in the discussion part. And here we did not list the changes but tracked the changes in the revised paper.

Point 4: “Concluding Thought

This was a strong paper and thorough process to get to these findings. I do not see any significant changes to be made.”

Response 4: We are grateful for the suggestion. Based on our findings, we have made an effort to improve this paper’ practical applications and conclusion part. line 649-666, line 688-692.